# Spontaneous Transvaginal Small Bowel Evisceration After Laparoscopic Hysterectomy

**DOI:** 10.3390/diagnostics14222498

**Published:** 2024-11-08

**Authors:** Ismini Kountouri, Amyntas Giotas, Christos Gkogkos, Ioannis Katsarelas, Panagiotis Nachopoulos, Afroditi Faseki, Dimitrios Chatzinas, Alexandra Panagiotou, Athanasios Polychronidis, Mohammad Husamieh, Periklis Dimasis, Nikolaos Gkiatas, Dimitra Manolakaki, Miltiadis Chandolias

**Affiliations:** 1Department of General Surgery, General Hospital of Katerini, 60100 Pieria, Greece; giannis24katsarelas@gmail.com (I.K.); panosnacho@gmail.com (P.N.); afroditifaseki@gmail.com (A.F.); jimmys055@gmail.com (D.C.); alexandrapanayiotou.larissa98@gmail.com (A.P.); thanospolychronidis@gmail.com (A.P.); 3bbadi1997@gmail.com (M.H.); dimasis@yahoo.com (P.D.); nikgiat71@gmail.com (N.G.); dimanolakaki@gmail.com (D.M.); miltoshandolias@gmail.com (M.C.); 2Gynecology and Obstetrics Department, General Hospital of Katerini, 60100 Pieria, Greece; ammag10@live.com (A.G.); akisgogos@yahoo.gr (C.G.)

**Keywords:** transvaginal evisceration, acute abdomen, total hysterectomy complications

## Abstract

Vaginal cuff dehiscence can be a rare complication of total hysterectomy, with an estimated prevalence of 0.032% to 1.25% and a high mortality rate of 6 to 10%. Dehiscence is also reported in cases following total laparoscopic hysterectomy, with a prevalence of 0.87%. This case report details the emergency management of a 59-year-old female who complained of abdominal and pelvic pain and the feeling of a foreign body in her vagina. The patient reported a history of laparoscopic total hysterectomy 6 months prior to presenting at the Emergency Department. A clinical examination revealed small bowel loops protruding through the vagina. The patient underwent exploratory laparotomy through a Pfannenstiel incision, and the terminal ileum was found prolapsing through the vaginal cuff. The bowel loops were identified as viable and the vagina was sutured. The patient had an unremarked post operative course. This case report showcases that in patients with transvaginal evisceration, immediate surgical management is crucial in order to avoid serious life threatening complications, and both surgeons and gynecologists should remain vigilant regarding this pathology.

A 59-year-old female presented at the Emergency Department of General Hospital of Katerini, in Greece, complaining of acute abdominal and pelvic pain and a feeling of a foreign body in her vagina, which had started three hours prior. The patient reported a history of a laparoscopic total hysterectomy, where the peritoneum was closed over the vaginal vault, 6 months prior to presenting at the Emergency Department, due to endometrial cancer, with no history of chemotherapy or radiotherapy. She also reported that she entered her menopausal status 9 years prior and had two normal vaginal deliveries at the ages of 30 and 32. A clinical examination with a speculum revealed small bowel loops protruding through the vagina (Figure 1).

A computer tomography scan revealed the existence of small bowel loops in the pelvis with free air bubbles surrounding them (Figure 2). The blood results on admission showed lactate of 1.4 mmol/L, C-reactive protein (CRP) 25 mg/L, hemoglobin (Hb) 10.9 g/L, and white blood cell count (WBC) 18.3 × 10^9^/L.

The decision for surgical management was made, and both a surgeon and a gynecologist participated in the operation. The patient underwent exploratory laparotomy through a Pfannenstiel incision. Upon laparotomy, the terminal ileum was found prolapsing through the vagina. The bowel loops were identified as viable and no bowel resection was required. The vagina was sutured, and the vaginal wall defect was repaired. The patient had an unremarked post operative course. She had no incidence of relapse during a three-year observation in our clinic.

Transvaginal small bowel evisceration is a rare surgical emergency, first reported in 1864, that requires immediate diagnosis and surgical treatment to avoid serious and life threatening complications [1]. Fewer than 100 cases have been reported in the literature, making this condition an extremely rare surgical emergency [1,2]. This condition is more frequent in postmenopausal women [1,2], while other risk populations include vaginal surgery cases, multiparae, and women of older age. This rare pathology can be a result of vaginal trauma that can be induced by coitus, misuse of obstetric instruments, or insertion of foreign bodies into the vagina [3]. Transvaginal evisceration generally has a higher incidence in postmenopausal women because of the thinner, scarred, foreshortened, and of diminished vascularity vaginal wall that makes it more susceptible to rupture [3]. Many factors such as increased abdominal pressure, vaginal ulceration due to severe atrophy, and straining at stool can be the cause of rupture of this sensitive postmenopausal vaginal wall [3].

The terminal ileum is usually the organ that protrudes through the vagina, but other organs such as the omentum, the colon, the fallopian tubes, and the appendix have also been reported to be involved [4]. The most common symptoms include the sense of tissue protruding through the vagina, vaginal pain, bleeding, and discharge [4]. This pathology has also be reported with a dramatic presentation in which large loops of small bowel prolapse through the vagina [4].

Vaginal cuff dehiscence is a rare occurrence after hysterectomy, with an estimated prevalence of 0.032% to 1.25% [1]. Dehiscence is reported to be more common after robotic-assisted total laparoscopic hysterectomy (2.33%) than total laparoscopic hysterectomy (0.87%), while the lowest rate is reported after complete vaginal hysterectomy [1].

A transvaginal bowel evisceration can have severe complications and has a mortality rate reaching up to 6–10%, mainly as a result of septicemia and thromboembolism [5]. Other serious complications are bowel infarction, infection, ileus, and deep vein thrombosis [5]. This pathology is a surgical emergency [1,2,3,5], and immediate management requires initial stabilization of the patient with intravenous fluid administration, cleaning and packing the bowel that is protruding through the vagina with moist saline sterile towels, early prophylactic antibiotic administration, and immediate surgical management in order to preserve the bowel viability and minimize the associated morbidity and mortality of this pathology [5].

In cases like ours where the small bowel protrudes through the vagina, surgical management requires a thorough examination of the bowel in order to decide if the loops are viable and when necessary to perform a resection of any ischemic and non-viable tissue [6]. Both vaginal and abdominal approaches have been reported with good outcomes for the patients, as well as a combined laparoscopic and vaginal procedure, which is only indicated in patients with viable incarcerated bowel that does not require any resection [2,5,6,7,8]. Laparotomy is widely accepted as the method of choice, because it allows the surgeon to perform a thorough inspection of the peritoneal cavity and perform a throughout peritoneal lavage. The transvaginal approach is more suitable for patients with a viable bowel and no signs of peritonitis, as this approach does not allow a thorough examination of the abdominal cavity [2]. In any case, any surgical treatment should be individualized for each patient [6].

In 1996, Kowalski et al. suggested specific surgical approaches in order to avoid vaginal evisceration. These include a surgical restoration of the normal vaginal axis, an anastomosis of the stumps of the supporting ligaments of the pelvis to the angles of the vagina for better support, a preservation of as much of vaginal length as possible, and an estrogen application for better vaginal integrity [2,9].

In a randomized trial in 2018, Uccela et al. concluded that laparoscopic closure of the vaginal cuff at the end of total laparoscopic hysterectomy is associated with a significant reduction in vaginal dehiscence, any cuff complication, vaginal bleeding, vaginal cuff hematoma, postoperative infection, need for vaginal resuture, and reintervention [10].

Through our case report, we intend to raise vigilance to both surgeons and gynecologists regarding this surgical emergency. As immediate surgical management is crucial in order to avoid the serious life threatening complications, in cases of pelvic and abdominal pain with or without discharge through the vagina, this rare pathology should be considered, in order to avoid visceral necrosis and peritonitis. 

## Figures and Tables

**Figure 1 diagnostics-14-02498-f001:**
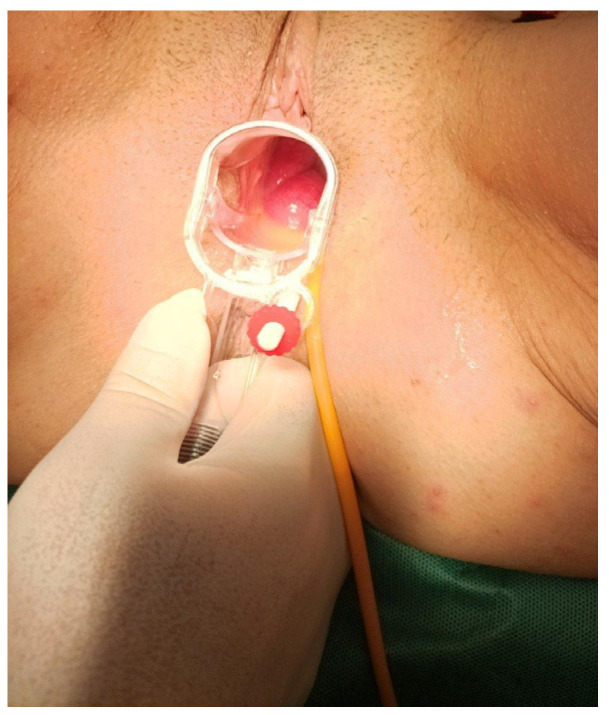
The small bowel loops protruding through the vagina identified upon clinical examination using a speculum.

**Figure 2 diagnostics-14-02498-f002:**
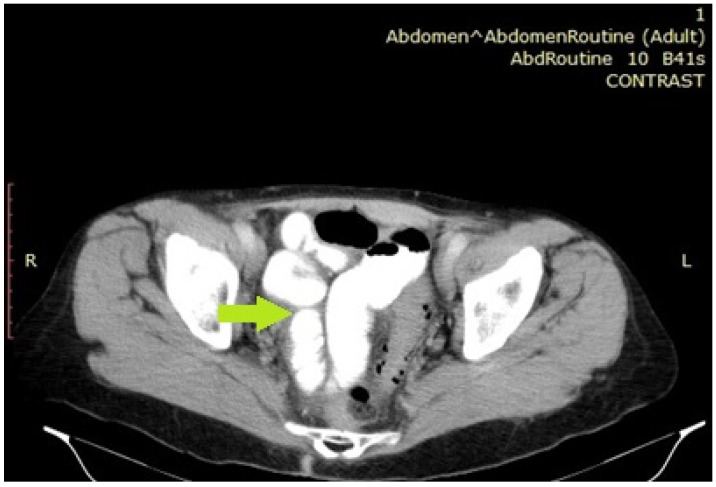
A computed tomography scan revealing the presence of small bowel loops in the pelvis with free air bubbles surrounding them. The small bowel loops are indicated by a green arrow.

## Data Availability

All relevant data are within the manuscript.

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
