# Peer review of "Spontaneous Transvaginal Small Bowel Evisceration After Laparoscopic Hysterectomy"

_diagnostics, 2024, doi:10.3390/diagnostics14222498_

Round 1
Reviewer 1 Report
Comments and Suggestions for Authors
This paper is an interesting presentation of a rare complication of abdominal hysterectomy. My queries are 1) Is it presented as a case report or simply as an
Image report as cited in the journal? If it is a case report it is not well presented. As an image report it should be summarised to a page so as to avoid the several controversial and incoherent arguments on the transvaginal, laparoscopic or open approaches in this complication. you should simply state why the index approach was taken.
2) As it is a rare complication, the authors should mention the type of malignancy incurred by the patient, her menopausal status and obstetric history, and if the peritoneum was closed over the vaginal vault at the initial operation.
3) It is 3 years since the incidence occurred and that delay may have jeopardized the recollection of your relevant data.
Comments on the Quality of English LanguageThe sentences are too long and not coherent especially in the last paragraph. I would advise you get English writers' support.
Author Response
Question 1:
1) Is it presented as a case report or simply as an Image report as cited in the journal? If it is a case report it is not well presented. As an image report it should be summarised to a page so as to avoid the several controversial and incoherent arguments on the transvaginal, laparoscopic or open approaches in this complication. you should simply state why the index approach was taken.
Response 1: Dear Reviewer,
This is a paper submitted as an Interesting Images article. We will revise and add more information regarding why the index approach was taken. With our case we indend to raise vigilance regarding the emergency surgical management of such cases and so we find it crucial to present all the available surgical treatment options.
2) As it is a rare complication, the authors should mention the type of malignancy incurred by the patient, her menopausal status and obstetric history, and if the peritoneum was closed over the vaginal vault at the initial operation.
Response 2 : We revised the text and added the required information.
3) It is 3 years since the incidence occurred and that delay may have jeopardized the recollection of your relevant data.
Response 3 : We observe the patient regularly in our Department
Reviewer 2 Report
Comments and Suggestions for Authors
The case report is well described and clear but I would like to suggest revision:
Authors well stated that this condition is "surgical emergency", thus I would stress the fact that both gynecologist and general surgeon should be involve in case management: the idea is to address the bowel viability (including option for resection) and reconstruction of vaginal wall.
Sometimes, omentum or other tissue should be used to cover the stump. Please advise.
Picture 2 (CT) is not express the pathology. It should be clear for any physician - at this version it doesn't.
I would suggest to stress the technical points and lessons learned.
This paper submitted to "interesting image" - unfortunately presented images are not "interesting" enough.
Author Response
Question 1 Αuthors well stated that this condition is "surgical emergency", thus I would stress the fact that both gynecologist and general surgeon should be involve in case management: the idea is to address the bowel viability (including option for resection) and reconstruction of vaginal wall.
Response 1 : The operation was performed by a gynecologist and a general surgeon. Both specialties were involved
Question 2 Sometimes, omentum or other tissue should be used to cover the stump. Please advise.
Response 2: The vagina was sutured and the vaginal wall defect was repaired. No omentum or other tissue was ussed to close the stump.
Question 3 : Picture 2 (CT) is not express the pathology. It should be clear for any physician - at this version it doesn't.
Response 3 Dear reviewer in Picture 2 you can see the small bowel loops prolapsing in the lesser pelvis , at close proximity with the sigmoid colon.
Question 4 I would suggest to stress the technical points and lessons learned.
Response 4
Through our case report we intend to raise vigilance to both surgeons and gynecologists regarding this surgical emergency. As immediate surgical management is crucial in order to avoid the serious life threatening complications, in cases of pelvic and abdominal pain with or without discharge through the vagina, especially in patients with previous gynecological operations this rare pathology should be considered, in order to avoid visceral necrosis and peritonitis.
Question 5 This paper submitted to "interesting image" - unfortunately presented images are not "interesting" enough.
Response 5 : This was the clinical image of the patient with small bowel loops prolapsing through the vaginal cuff.
Reviewer 3 Report
Comments and Suggestions for Authors
Dear authors, although your case report is exceptional and deserving of publication, your manuscript contains scholarly, grammatical, and orthographic errors.
Detailed remarks:
The whole manuscript contains multiple spelling, punctuation, orthographic, and grammar errors. It requires review by a native English speaker.
The title is incorrect as it does not match the case description. The case describes a patient who underwent a laparoscopic total hysterectomy; however, the title indicates a patient with a history of undergoing an abdominal hysterectomy. The title should reflect the type of hysterectomy performed. The title is the most read part of an article; therefore, it must attract the reader’s attention.
The abstract contains misconceptions. The text indicates “Clinical examination revealed small bowel loops protruding through the vaginal introitus.” However, Figure 1 depicted the small bowel loops protruding through the vagina, rather than the introitus, which was only identified using a speculum. The other mistake is “…terminal ileum was found prolapsing through a vaginal wall defect.” This is incorrect; the small bowel was protruding through the vaginal cuff. The abstract should be instructive and brief on the study, validate noteworthy information, and pay attention to significant findings and conclusions. Moreover, the abstract is a critical part of the article that requires the author's precision to capture the reader's attention.
Please ensure the case report specifies whether a trigger factor for bowel evisceration was identified. For how long has the patient been experiencing acute abdominal and pelvic pain, along with a sensation of a foreign body in her vagina? Figure 2 needs arrows or indicators to explain the image characteristics. Case reports should include important positive and negative findings from the patient's history, physical examination, and any investigations conducted.
I discovered evidence of plagiarism. Some parts of the discussion exhibit a plagiarism rate ranging from 6% to 18%. Discussion is poor. Start by summarizing your main findings. Then, use your arguments to demonstrate the importance of your work. Keep in mind that you should not just repeat the results but instead offer a thorough interpretation of the data to explain the significance of your findings.
Comments on the Quality of English LanguageThe whole manuscript contains multiple spelling, punctuation, orthographic, and grammar errors. It requires review by a native English speaker.
Author Response
Dear authors, although your case report is exceptional and deserving of publication, your manuscript contains scholarly, grammatical, and orthographic errors.
Detailed remarks:
The whole manuscript contains multiple spelling, punctuation, orthographic, and grammar errors. It requires review by a native English speaker.
Comment 1 The title is incorrect as it does not match the case description. The case describes a patient who underwent a laparoscopic total hysterectomy; however, the title indicates a patient with a history of undergoing an abdominal hysterectomy. The title should reflect the type of hysterectomy performed. The title is the most read part of an article; therefore, it must attract the reader’s attention.
Response 1 We revised the title . Thank you for your comment.
Comment 2: The abstract contains misconceptions. The text indicates “Clinical examination revealed small bowel loops protruding through the vaginal introitus.” However, Figure 1 depicted the small bowel loops protruding through the vagina, rather than the introitus, which was only identified using a speculum.
Response 2 We revised the text
Comment 3: The other mistake is “…terminal ileum was found prolapsing through a vaginal wall defect.” This is incorrect; the small bowel was protruding through the vaginal cuff. The abstract should be instructive and brief on the study, validate noteworthy information, and pay attention to significant findings and conclusions. Moreover, the abstract is a critical part of the article that requires the author's precision to capture the reader's attention.
Response 3 : We revised the text
Comment 4: Please ensure the case report specifies whether a trigger factor for bowel evisceration was identified. For how long has the patient been experiencing acute abdominal and pelvic pain, along with a sensation of a foreign body in her vagina?
Response 4 : We added the information.
Comment 5 : Figure 2 needs arrows or indicators to explain the image characteristics.
Response 5: We added the arrows.
Comment 6 : Case reports should include important positive and negative findings from the patient's history, physical examination, and any investigations conducted.
Response 6: We revised the text : A 59 year old female presented at the Emergency Department of General Hospital of Katerini, in Greece, complaining of acute abdominal and pelvic pain and a feeling of a foreign body in her vagina, started three hours ago. The patient reported a history of a laparoscopic total hysterectomy, where the peritoneum was closed over the vaginal vault, 6 months prior due to endometrial cancer with no history of chemotherapy or radiotherapy.. She also reported that she entered her menopausal status 9 years ago and had 2 normal vaginal deliveries of her 2 children at the age of 30 and 32. Clinical examination with a speculum revealed small bowel loops protruding through the vagina( Figure 1).
Comment 7 : I discovered evidence of plagiarism. Some parts of the discussion exhibit a plagiarism rate ranging from 6% to 18%.
Response 7: We will revise the text.
Comment 8 : Discussion is poor. Start by summarizing your main findings. Then, use your arguments to demonstrate the importance of your work. Keep in mind that you should not just repeat the results but instead offer a thorough interpretation of the data to explain the significance of your findings.
Response 8 Dear reviewer this is an interesting images case and so there is no discussion section. We mainly present other cases like ours to indicate that this is indeed a surgical imergency and that other cases are reported in the literature.
Round 2
Reviewer 1 Report
Comments and Suggestions for Authors
the corrected version is acceptable.
Reviewer 2 Report
Comments and Suggestions for Authors
Thank you for addressing all comments.
Once again, this case was submitted under the "interesting images" topic.
Picture 2 could be presented in frontal view.
There are many case reports like yours. What make it unique?
Reviewer 3 Report
Comments and Suggestions for Authors
Dear authors, although your case report is exceptional and deserving of publication, your manuscript persists with some minor orthographic flaws in the text.
Detailed remarks:
Patient age is wrongly written (line 25: A 59 year old female…); it must be: A 59-year-old female….
Comments on the Quality of English LanguageThe text persists with some minor orthographic flaws in the text.